# POST TRAINING IN DEEP LEARNING

## ABSTRACT

One of the main challenges of deep learning methods is the choice of an appropriate training strategy. In particular, additional steps, such as unsupervised pretraining, have been shown to greatly improve the performances of deep structures. In this article, we propose an extra training step, called post-training, which only optimizes the last layer of the network. We show that this procedure can be analyzed in the context of kernel theory, with the first layers computing an embedding of the data and the last layer a statistical model to solve the task based on this embedding. This step makes sure that the embedding, or representation, of the data is used in the best possible way for the considered task. This idea is then tested on multiple architectures with various data sets, showing that it consistently provides a boost in performance.

## 1 TRAINING NEURAL NETWORKS

One of the main challenges of the deep learning methods is to efficiently solve the highly complex and non-convex optimization problem involved in the training step. Many parameters influence the performances of trained networks, and small mistakes can drive the algorithm into a sub-optimal local minimum, resulting into poor performances (Bengio & LeCun, 2007). Consequently, the choice of an appropriate training strategy is critical to the usage of deep learning models.

The most common approach to train deep networks is to use the stochastic gradient descent (SGD) algorithm. This method selects a few points in the training set, called a batch, and compute the gradient of a cost function relatively to all the layers parameter. The gradient is then used to update the weights of all layers. Empirically, this method converges most of the time to a local minimum of the cost function which have good generalization properties. The stochastic updates estimate the gradient of the error on the input distribution, and several works proposed to use variance reduction technique such as Adagrap (Duchi et al., 2011), RMSprop (Hinton et al., 2012) or Adam (Kingma & Ba, 2015), to achieve faster convergence.

While these algorithms converge to a local minima, this minima is often influenced by the properties of the initialization used for the network weights. A frequently used approach to find a good starting point is to use pre-training (Larochelle et al., 2007; Hinton et al., 2006; Hinton & Salakhutdinov, 2006). This method iteratively constructs each layer using unsupervised learning to capture the information from the data. The network is then fine-tuned using SGD to solve the task at hand. Pre-training strategies have been applied successfully to many applications, such as classification tasks (Bengio & LeCun, 2007; Poultney et al., 2006), regression (Hinton & Salakhutdinov, 2008), robotics (Hadsell et al., 2008) or information retrieval (Salakhutdinov & Hinton, 2009). The influence of different pre-training strategies over the different layers has been thoroughly studied in Larochelle et al. (2009). In addition to improving the training strategies, these works also shed light onto the role of the different layers (Erhan et al., 2010; Montavon et al., 2011). The first layers of a deep neural network, qualified as *general*, tend to learn feature extractors which can be reused in other architectures, independently of the solved task. Meanwhile, the last layers of the network are much more dependent of the task and data set, and are said to be *specific*.

Deep Learning generally achieves better results than shallow structures, but the later are generally easier to train and more stable. For convex models such as logistic regression, the training problem is also convex when the data representation is fixed. The separation between the representation and the model learning is a key ingredient for the model stability. When the representation is learned simultaneously, for instance with dictionary learning or with EM algorithms, the problem often

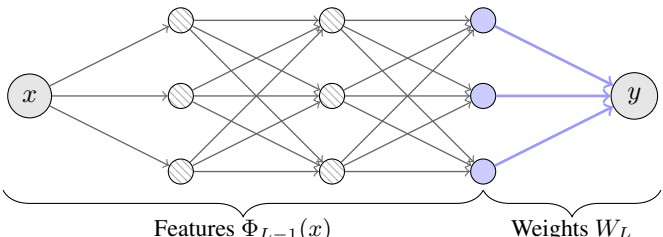

Features $\Phi_{L-1}(x)$       Weights $W_L$

Figure 1: Illustration of post-training applied to a neural network. During the post-training, only the weights of the blue edges are updated. The blue nodes can be seen as the embedding of $x$ in the feature space $\mathcal{X}_L$.

become non-convex. But this coupling between the representation and the model is critical for end-to-end models. For instance, Hinton et al. (2006) showed that for networks trained using pre-training, the fine-tuning step – where all the layers are trained together – improves the performances of the network. This shows the importance of the adaptation of the representation to the task in end-to-end models.

Our contribution in this chapter is an additional training step which improves the use of the representation learned by the network to solve the considered task. This new step is called *post-training*. It is based on the idea of separating representation learning and statistical analysis and it should be used after the training of the network. In this step, only the specific layers are trained. Since the general layers – which encode the data representation – are fixed, this step focuses on finding the best usage of the learned representation to solve the desired task. In particular, we chose to study the case where only the last layer is trained during the post-training, as this layer is the most specific one (Yosinski et al., 2014). In this setting, learning the weights of the last layer corresponds to learning the weights for the kernel associated to the feature map given by the previous layers. The post-training scheme can thus be interpreted in light of different results from kernel theory. To summarize our contributions:

- We introduce a post-training step, where all layers except the last one are frozen. This method can be applied after any traditional training scheme for deep networks. Note that this step does not replace the end-to-end training, which co-adapts the last layer representation with the solver weights, but it makes sure that this representation is used in the most efficient way for the given task.

- We show that this post-training step is easy to use, that it can be effortlessly added to most learning strategies, and that it is computationally inexpensive.

- We highlight the link existing between this method and the kernel techniques. We also show numerically that the previous layers can be used as a kernel map when the problem is small enough.

- We experimentally show that the post-training does not overfit and often produces improvement for various architectures and data sets.

The rest of this article is organized as follows: Section 2 introduces the post-training step and discusses its relation with kernel methods. Section 4 presents our numerical experiments with multiple neural network architectures and data sets and Section 5 discusses these results.

## 2   POST-TRAINING

In this section, we consider a feedforward neural network with $L$ layers, where $\mathcal{X}_1, \ldots, \mathcal{X}_L$ denote the input space of the different layers, typically $\mathbb{R}^{d_l}$ with $d_l > 0$ and $\mathcal{Y} = \mathcal{X}_{L+1}$ the output space of our network. Let $\phi_l : \mathcal{X}_l \mapsto \mathcal{X}_{l+1}$ be the applications which respectively compute the output of the $l$-th layer of the network, for $1 \leq l \leq L$, using the output of the $l-1$-th layer and $\Phi_L = \phi_L \circ \cdots \circ \phi_1$ be the mapping of the full network from $\mathcal{X}_1$ to $\mathcal{Y}$. Also, for each layer $l$, we denote $\boldsymbol{W}_l$ its weights matrix and $\psi_l$ its activation function.

The training of our network is done using a convex and continuous loss function $\ell : \mathcal{Y} \times \mathcal{Y} \mapsto \mathbb{R}^+$. The objective of the neural network training is to find weights parametrizing $\Phi_L$ that solves the following problem:

$$\min_{\Phi_L} \mathbb{E}_{(x,y) \sim \mathcal{P}} \left[ \ell \left( \Phi_L(x), y \right) \right] . \tag{1}$$

for a certain input distribution $\mathcal{P}$ in $(\mathcal{X}_1, \mathcal{Y})$. The training set is $\mathcal{D} = (x_i, y_i)_{i=1}^N$, drawn from this input distribution.

Using these notations, the training objective (1) can then be rewritten

$$\min_{\Phi_{L-1}, \boldsymbol{W}_L} \mathbb{E}_{(x,y) \sim \mathcal{P}} \left[ \ell \left( \psi_L \left( \boldsymbol{W}_L \Phi_{L-1}(x) \right), y \right) \right] . \tag{2}$$

This reformulation highlights the special role of the last layer in our network compared to the others. When $\Phi_{L-1}$ is fixed, the problem of finding $\boldsymbol{W}_L$ is simple for several popular choices of activation function $\psi_L$ and loss $\ell$. For instance, when the activation function $\psi_L$ is the `softmax` function and the loss $\ell$ is the `cross entropy`, (2) is a multinomial logistic regression. In this case, training the last layer is equivalent to a regression of the labels $y$ using the embedding of the data $x$ in $\mathcal{X}_L$ by the mapping $\Phi_{L-1}$. Since the problem is convex in $\boldsymbol{W}_L$ (see Appendix A), classical optimization techniques can efficiently produce an accurate approximation of the optimal weights $\boldsymbol{W}_L$ – and this optimization given the mapping $\Phi_{L-1}$ is the idea behind post-training.

Indeed, during the regular training, the network tries to simultaneously learn suitable representation for the data in the space $\mathcal{X}_L$ through its $L-1$ first layer and the best use of this representation with $\boldsymbol{W}_L$. This joint minimization is a strongly non-convex problem, therefore resulting in a potentially sub-optimal usage of the learned data representation.

The post-training is an additional step of learning which takes place after the regular training and proceeds as follows :

1. **Regular training:** This step aims to obtain interesting features to solve the initial problem, as in any usual deep learning training. Any training strategy can be applied to the network, optimizing the empirical loss

$$\operatorname*{argmin}_{\Phi_L} \frac{1}{N} \sum_{i=1}^N \ell \left( \Phi_L(x_i), y_i \right) . \tag{3}$$

   The stochastic gradient descent explores the parameter space and provides a solution for $\Phi_{L-1}$ and $\boldsymbol{W}_L$. This step is non restrictive: any type of training strategy can be used here, including gradient bias reduction techniques, such as Adagrad (Duchi et al., 2011), or regularization strategies, for instance using Dropout (Dahl et al., 2013). Similarly, any type of stopping criterion can be used here. The training might last for a fixed number of epochs, or can stop after using early stopping (Morgan & Bourlard, 1990). Different combinations of training strategies and stopping criterion are tested in Section 4.

2. **Post-training:** During this step, the first $L-1$ layers are fixed and only the last layer of the network, $\phi_L$, is trained by minimizing over $\boldsymbol{W}_L$ the following problem

$$\operatorname*{argmin}_{\boldsymbol{W}_L} \frac{1}{N} \sum_{i=1}^N \tilde{\ell} \left( \Phi_{L-1}(x_i) \boldsymbol{W}_L^\mathsf{T}, y_i \right) + \lambda \|\boldsymbol{W}_L\|_2^2 , \tag{4}$$

   where $\tilde{\ell}(x, y) := \ell(\psi_L(x), y)$. This extra learning step uses the mapping $\Phi_{L-1}$ as an embedding of the data in $\mathcal{X}_L$ and learn the best linear predictor in this space. This optimization problem takes place in a significantly lower dimensional space and since there is no need for back propagation, this step is computationally faster. To reduce the risk of overfitting with this step, a $\ell_2$-regularization is added. Figure 1 illustrates the post-training step.

We would like to emphasize the importance of the $\ell_2$-regularization used during the post-training (4). This regularization is added regardless of the one used in the regular training, and for all the network architectures. The extra term improves the strong convexity of the minimization problem,

making post-training more efficient, and promotes the generalization of the model. The choice of the $\ell_2$-regularization is motivated from the comparison with the kernel framework discussed in Section 3 and from our experimental results.

**Remark 1** (Dropout.). *It is important to note that Dropout should not be applied on the previous layers of the network during the post-training, as it would lead to changes in the feature function $\Phi_{L-1}$.*

## 3 LINK WITH KERNELS

In this section, we show that for the case where $\mathcal{X}_L = \mathbb{R}^{d_L}$ for some $d_L > 0$ and $\mathcal{X}_{L+1} = \mathbb{R}$, $\boldsymbol{W}_L^*$ can be approximated using kernel methods. We define the kernel $k$ as follows,

$$k : \mathcal{X}_1 \times \mathcal{X}_1 \mapsto \mathbb{R}$$

$$(x_1, x_2) \to \left\langle \Phi_{L-1}(x_1), \Phi_{L-1}(x_2) \right\rangle .$$

Then $k$ is the kernel associated with the feature function $\Phi_{L-1}$. It is easy to see that this kernel is continuous positive definite and that for $\boldsymbol{W} \in \mathbb{R}^{d_L}$, the function

$$g_{\boldsymbol{W}} : \mathcal{X}_1 \mapsto \mathcal{X}_{L+1}$$

$$x \to \left\langle \Phi_{L-1}(x), \boldsymbol{W} \right\rangle \tag{5}$$

belongs by construction to the Reproducing Kernel Hilbert Space (RKHS) $\mathcal{H}_k$ generated by $k$. The post-training problem (4) is therefore related to the problem posed in the RKHS space $\mathcal{H}_k$, defined by

$$g^* = \underset{g \in \mathcal{H}_k}{\operatorname{argmin}} \frac{1}{N} \sum_{i=1}^N \tilde{\ell}\Big(g(x_i), y_i\Big) + \lambda \|g\|_{\mathcal{H}_k}^2 ,$$

This problem is classic for the kernel methods. With mild hypothesis on $\tilde{\ell}$, the generalized representer theorem can be applied (Schölkopf et al., 2001). As a consequence, there exists $\alpha^* \in \mathbb{R}^N$ such that

$$g^* := \underset{g \in \mathcal{H}_k}{\operatorname{argmin}} \frac{1}{N} \sum_{i=1}^N \tilde{\ell}\Big(g(x_i), y_i\Big) + \lambda \|g\|_{\mathcal{H}_k}^2$$

$$= \sum_{i=1}^N \alpha_i^* k(X_i, \cdot) = \sum_{i=1}^N \left\langle \alpha_i^* \Phi_{L-1}\left(x_i\right), \Phi_{L-1}(\cdot) \right\rangle. \tag{6}$$

Rewriting (6) with $g^*$ of the form (5), we have that $g^* = g_{\boldsymbol{W}^*}$, with

$$\boldsymbol{W}^* = \sum_{i=1}^N \alpha_i^* \Phi_{L-1}\left(x_i\right) \text{`.} \tag{7}$$

We emphasize that $\boldsymbol{W}^*$ gives the optimal solution for the problem (6) and should not be confused with $\boldsymbol{W}_L^*$, the optimum of (4). However, the two problems differ only in their regularization, which are closely related (see the next paragraph). Thus $\boldsymbol{W}^*$ can thus be seen as an approximation of the optimal value $\boldsymbol{W}_L^*$. It is worth noting that in our experiments, $\boldsymbol{W}^*$ appears to be a nearly optimal estimator of $\boldsymbol{W}_L^*$ (see Subsection 4.3).

**Relation between $\|\cdot\|_{\mathcal{H}}$ and $\|\cdot\|_2$.** The problems (6) and (4) only differ in the choice of the regularization norm. By definition of the RKHS norm, we have

$$\|g_W\|_{\mathcal{H}} = \inf\left\{ \|v\|_2 \Big/ \quad \forall x \in \mathcal{X}_1, \quad \langle v, \Phi_{L-1}(x) \rangle = g_W(x) \right\} .$$

Consequently, we have that $\|g_W\|_{\mathcal{H}} \leq \|W\|_2$, with equality when $\text{Vect}(\Phi_{L-1}(\mathcal{X}_1))$ spans the entire space $\mathcal{X}_L$. In this case, the norm induced by the RKHS is equal to the $\ell_2$-norm. This is generally the case, as the input space is usually in a far higher dimensional space than the embedding space, and since the neural network structure generally enforces the independence of the features. Therefore, while both norms can be used in (4), we chose to use the $\ell_2$-norm for all our experiments as it is easier to compute than the RKHS norm.

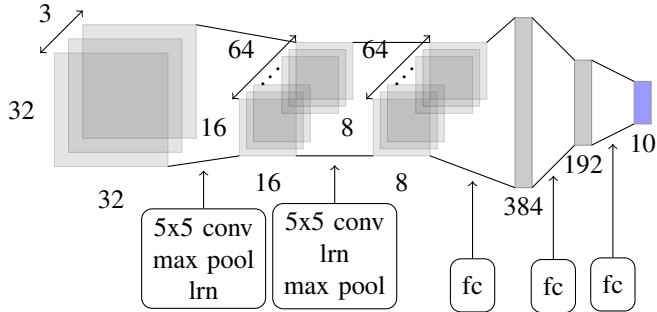

Figure 2: Illustration of the neural network structure used for CIFAR-10. The last layer, represented in blue, is the one trained during the post-training. The layers are composed with classical layers: 5x5 convolutional layers (5x5 conv), max pooling activation (max pool), local response normalization (lrn) and fully connected linear layers (fc).

**Close-form Solution.** In the particular case where $\ell(y_1, y_2) = \|y_1 - y_2\|^2$ and $f(x) = x$, (6) can be reduced to a classical Kernel Ridge Regression problem. In this setting, $W^*$ can be computed by combining (7) and

$$\alpha^* = \left( \Phi_{L-1}(\mathcal{D})^\mathsf{T} \Phi_{L-1}(\mathcal{D}) + \lambda \boldsymbol{I}_N \right)^{-1} Y \,, \tag{8}$$

where $\Phi_{L-1}(\mathcal{D}) = \left[ \Phi_{L-1}(x_1), \dots \Phi_{L-1}(x_N) \right]$ represents the matrix of the input data $\{x_1, \dots x_N\}$ embedded in $\mathcal{X}_L$, $Y$ is the matrix of the output data $\{y_1, \dots, y_N\}$ and $\boldsymbol{I}_N$ is the identity matrix in $\mathbb{R}^N$. This result is experimentally illustrated in Subsection 4.3. Although data sets are generally too large for (8) to be computed in practice, it is worth noting that some kernel methods, such as Random Features (Rahimi & Recht, 2007), can be applied to compute approximations of the optimal weights during the post-training.

**Multidimensional Output.** Most of the previously discussed results related to kernel theory hold for multidimensional output spaces, *i.e.* $\dim(\mathcal{X}_{L+1}) = d > 1$, using multitask or operator valued kernels (Kadri et al., 2015). Hence the previous remarks can be easily extended to multidimensional outputs, encouraging the use of post-training in most settings.

## 4 EXPERIMENTAL RESULTS

This section provides numerical arguments to study post-training and its influence on performances, over different data sets and network architectures. All the experiments were run using `python` and `Tensorflow`. The code to reproduce the figures is available online[1]. The results of all the experiments are discussed in depth in Section 5.

### 4.1 CONVOLUTIONAL NEURAL NETWORKS

The post-training method can be applied easily to feedforward convolutional neural network, used to solve a wide class of real world problems. To assert its performance, we apply it to three classic benchmark datsets: CIFAR10 (Krizhevsky, 2009), MNIST and FACES (Hinton & Salakhutdinov, 2006).

**CIFAR10.** This data set is composed of $60,000$ images $32 \times 32$, representing objects from $10$ classes. We use the default architecture proposed by `Tensorflow` for CIFAR10 in our experiments, based on the original architecture proposed by Krizhevsky (2009). It is composed of $5$ layers described in Figure 2. The first layers use various common tools such as local response normalization (lrn), max pooling and RELU activation. The last layer have a *softmax* activation function

---

[1]The code is available at `anonymous github repo`

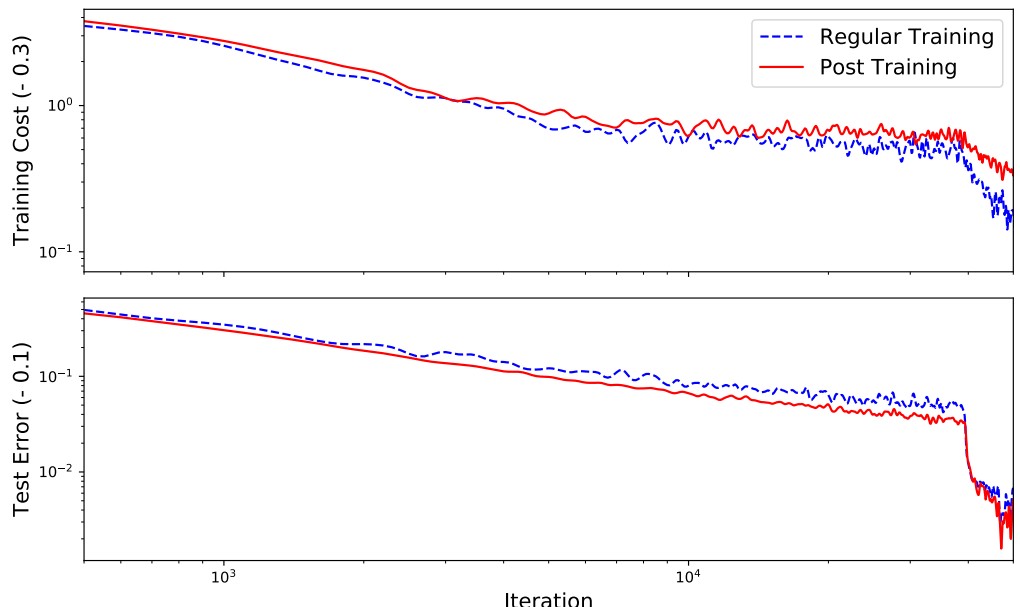

Figure 3: Evolution of the performances of the neural network on the CIFAR10 data set, (*dashed*) with the usual training and (*solid*) with the post-training phase. For the post-training, the value of the curve at iteration $q$ is the error for a network trained for $q - 100$ iterations with the regular training strategy and then trained for $100$ iterations with post-training. The *top* figure presents the classification error on the training set and the *bottom* figure displays the loss cost on the test set. The curves have been smoothed to increase readability.

and the chosen training loss was the cross entropy function. The network is trained for $90k$ iterations, with batches of size $128$, using stochastic gradient descent (SGD), dropout and an exponential weight decay for the learning rate. Figure 3 presents the performance of the network on the training and test sets for 2 different training strategies. The dashed line present the classic training with SGD, with performance evaluated every $100$ iterations and the solid line present the performance of the same network where the last $100$ iterations are done using post-training instead of regular training. To be clearer, the value of this curve at iteration $q$ is the error of the network, trained for $q - 100$ iterations with the regular training strategy, and then trained for $100$ iterations with post-training. The regularization parameter $\lambda$ for post-training is set to $1 \times 10^{-3}$.

The results show that while the training cost of the network mildly increases due to the use of post-training, this extra step improves the generalization of the solution. The gain is smaller at the end of the training as the network converges to a local minimum, but it is consistent. Also, it is interesting to note that the post-training iterations are $4\times$ faster than the classic iterations, due to their inexpensiveness.

**Additional Data Sets.** We also evaluate post-training on the MNIST data set ($65000$ images $27 \times 27$, with $55000$ for train and $10000$ for test; 10 classes) and the pre-processed FACES data set ($400$ images $64 \times 64$, from which $102400$ sub-images, $32 \times 32$, are extracted, with $92160$ for training and $10240$ for testing; 40 classes). For each data set, we train two different convolutional neural networks – to assert the influence of the complexity of the network over post-training:

- a small network, with one convolutional layer ($5 \times 5$ patches, 32 channels), one $2 \times 2$ max pooling layer, and one fully connected hidden layer with $512$ neurons,

- a large network, with one convolutional layer ($5 \times 5$ patches, 32 channels), one $2 \times 2$ max pooling layer, one convolutional layer ($5 \times 5$ patches, 64 channels), one $2 \times 2$ max pooling layer and one fully connected hidden layer with $1024$ neurons.

Table 1: Comparison of the performances (classification error) of different networks on different data sets, at different epochs, with or without post-training.

| Data set | Network | Iterations | Mean (Std) Error in % | Mean (Std) Error with post-training in % |
|---|---|---|---|---|
| FACES | Small | 5000 | 21,5 (10) | 19,1 (12) |
| | | 10000 | 20 (4) | 19 (3,5) |
| | | 20000 | 18 (0,9) | 16,5 (0,8) |
| | Large | 5000 | 25 (15) | 24 (15) |
| | | 10000 | 15 (5) | 12 (5) |
| | | 20000 | 11 (0,5) | 10 (0,5) |
| MNIST | Small | 1000 | 10.7 (1) | 9.2 (1,1) |
| | | 2000 | 7,5 (0,7) | 6,7 (0,6) |
| | | 5000 | 4,1 (0,2) | 3,9 (0,2) |
| | Large | 1000 | 9,1 (1,3) | 8,5 (1,4) |
| | | 2000 | 4,1 (0,2) | 3,5 (0,2) |
| | | 5000 | 1,1 (0,01) | 0,9 (0,01) |

We use dropout for the regularization, and set $\lambda = 1 \times 10^{-2}$. We compare the performance gain resulting of the application of post-training (100 iterations) at different epochs of each of these networks. The results are reported in Table 1.

As seen in Table 1, post-training improves the test performance of the networks with as little as 100 iterations – which is negligible compared to the time required to train the network. While the improvement varies depending on the complexity of the network, of the data set, and of the time spent training the network, it is important to remark that it always provides an improvement.

## 4.2 RECURRENT NEURAL NETWORK

While the kernel framework developed in Section 2 does not apply directly to Recurrent Neural Network, the idea of post-training can still be applied. In this experiment, we test the performances of post-training on Long Short-Term Memory-based networks (LSTM), using PTB data set (Marcus et al., 1993).

**Penn Tree Bank (PTB).** This data set is composed of $929k$ training words and $82k$ test word, with a 10000 words vocabulary. We train a recurrent neural network to predict the next word given the word history. We use the architecture proposed by Zaremba et al. (2014), composed of 2 layers of 1500 LSTM units with tanh activation, followed by a fully connected softmax layer. The network is trained to minimize the average per-word perplexity for 100 epochs, with batches of size 20, using gradient descent, an exponential weight decay for the learning rate, and dropout for regularization. The performances of the network after each epoch are compared to the results obtained if the 100 last steps (i.e. 100 batches) are done using post-training. The regularization parameter for post-training, $\lambda$, is set to $1 \times 10^{-3}$. The results are reported in Figure 4, which presents the evolution of the training and testing perplexity.

Similarly to the previous experiments, post-training improves the test performance of the networks, even after the network has converged.

## 4.3 OPTIMAL LAST LAYER FOR DEEP RIDGE REGRESSION

In this subsection we aim to empirically evaluate the close-form solution discussed in Section 2 for regression tasks. We set the activation function of the last layer to be the identity $f_L(x) = x$, and consider the loss function to be the least-squared error $\ell(x, y) = \|x - y\|_2^2$ in (1). In in each experiment, (8) and (7) are used to compute $W^*$ for the kernel learned after the regular training of

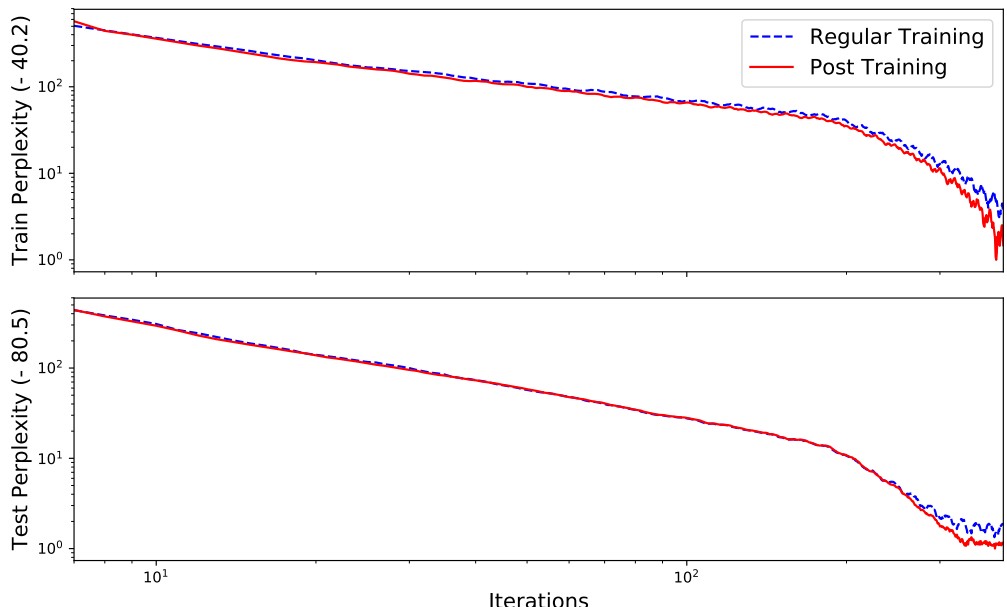

Figure 4: Evolution of the performances of the Recurrent network on the PTB data set. The *top* figure presents the train perplexity and the *bottom* figure displays the test perplexity. For the post-training, the value of the curve at iteration $q$ is the error for a network trained for $q - 100$ iterations with the regular training strategy and then trained for 100 iterations with post-training.

the neural network, which learn the embedding $\Phi_{L-1}$ and an estimate $W_L$ . In order to illustrate this result, and to compare the performances of the weights $W^*$ with respect to the weights $W_L$, learned either with usual learning strategies or with post-training, we train a neural network on two regression problems using a real and a synthetic data set. 70% of the data are used for training, and 30% for testing.

**Real Data Set Regression.**    For this experiment, we use the Parkinson Telemonitoring data set (Tsanas et al., 2010). The input consists in $5,875$ instances of 17 dimensional data, and the output are one dimensional real number. For this data set, a neural network made of two fully connected hidden layers of size 17 and 10 with respectively `tanh` and RELU activation, is trained for $250$, $500$ and $750$ iterations, with batches of size 50. The layer weights are all regularized with the $\ell_2$-norm and a fixed regularization parameter $\lambda = 10^{-3}$ . Then, starting from each of the trained networks, 200 iterations of post-training are used with the same regularization parameter $\lambda$ and the performances are compared to the closed-form solutions computed using (8) for each saved network. The results are presented in Table 2.

**Simulated Data Set Regression.**    For this experiment, we use a synthetic data set. The inputs were generated using a uniform distribution on $\left[0, 1\right]^{10}$. The outputs are computed as follows:

$$Y = \tanh(XW_1)W_2$$

where $W_1 \in \left[-1, 1\right]^{10 \times 5}$ and $W_2 \in \left[-1, 1\right]^5$ are randomly generated using a uniform law. In total, the data set is composed of $10,000$ pairs $(x_i, y_j)$. For this data set, a neural network with two fully connected hidden layers of size 10 with activation `tanh` for the first layer and RELU for the second layer is trained for $250, 500$ and $750$ iterations, with batches of size 50. We use the same protocol with 200 extra post-training iterations. The results are presented in Table 2.

For these two experiments, the post-training improves the performances toward these of the optimal solution, for several choices of stopping times. It is worth noting that the performance of the optimal solution is better when the first layers are not fully optimized with Parkinson Telemonitoring data set. This effect denotes an overfitting tendency with the full training, where the first layers become overly specified for the training set.

Table 2: Comparison of the performances (RMSE) of fully connected networks on different data sets, at different epochs, with or without post-training.

| Data set | Iterations | Error with classic training | Error with post-training | Error with optimal last layer |
|---|---|---|---|---|
| Parkinson | 250 | 0.832 | 0.434 | 0.119 |
| | 500 | 0.147 | 0.147 | 0.140 |
| | 750 | 0.134 | 0.132 | 0.131 |
| Simulated | 250 | 1.185 | 1.117 | 1.075 |
| | 500 | 0.533 | 0.450 | 0.447 |
| | 750 | 0.322 | 0.300 | 0.296 |

## 5 DISCUSSION

The experiments presented in Section 4 show that post-training improves the performances of all the networks considered – including recurrent, convolutional and fully connected networks. The gain is significant, regardless of the time at which the regular training is stopped and the post-training is done. In both the CIFAR10 and the PTB experiment, the gap between the losses with and without post-training is more pronounced if the training is stopped early, and tends to be smaller as the network converges to a better solution (see Figure 4 and Figure 3). The reduction of the gap between the test performances with and without post-training is made clear in Table 1. For the MNIST data set, with a small-size convolutional neural network, while the error rate drops by 1.5% when post-training is applied after 5000 iterations, this same error rate only drops by 0.2% when it is applied after 20000 iterations. This same observation can be done for the other results reported in Table 1. However, while the improvement is larger when the network did not fully converge prior to the post-training, it is still significant when the network has reached its minimum: for example in PTB the final test perplexity is 81.7 with post-training and 82.4 without; in CIFAR10 the errors are respectively 0.147 and 0.154.

If the networks are allowed to moderately overfit, for instance by training them with regular algorithm for a very large number of iterations, the advantage provided by post-training vanishes: for example in PTB the test perplexity after 2000 iterations (instead of 400) is 83.2 regardless of post-training. This is coherent with the intuition behind the post-training: after overfitting, the features learned by the network become less appropriate to the general problem, therefore their optimal usage obtained by post-training no longer provide an advantage.

It is important to note that the post-training computational cost is very low compared to the full training computations. For instance, in the CIFAR10 experiment, each iteration for post-training is $4\times$ faster on the same GPU than an iteration using the full gradient. Also, in the different experiments, post-training produces a performance gap after using as little as 100 batches. There are multiple reasons behind this efficiency: first, the system reaches a local minimum relatively rapidly for post-training as the problem (4) has a small number of parameters compared to the dimensionality of the original optimization problem. Second, the iterations used for the resolution of (4) are computationally cheaper, as there is no need to chain high dimensional linear operations, contrarily to regular backpropagation used during the training phase. Finally, since the post-training optimization problem is generally convex, the optimization is guaranteed to converge rapidly to the optimal weights for the last layer.

Another interesting point is that there is no evidence that the post-training step leads to overfitting. In CIFAR10, the test error is improved by the use of post-training, although the training loss is similar. The other experiments do not show signs of overfitting either as the test error is mostly improved by our additional step. This stems from the fact that the post-training optimization is much simpler than the original problem as it lies in a small-dimensional space – which, combined with the added $\ell_2$-regularization, efficiently prevents overfitting. The regularization parameter $\lambda$ plays an important role in post-training. Setting $\lambda$ to be very large reduces the explanatory capacity of the

networks whereas if $\lambda$ is too small, the capacity can become too large and lead to overfitting. Overall, our experiments highlighted that the post-training produces significant results for any choice of $\lambda$ reasonably small (i.e $10^{-5} \leq \lambda \leq 10^{-2}$ ). This parameter is linked to the regularization parameter of the kernel methods, as stated in Section 3.

Overall, these results show that the post-training step can be applied to most trained networks, without prerequisites about how optimized they are since post-training does not degrade their performances, providing a consistent gain in performances for a very low additional computational cost.

In Subsection 4.3, numerical experiments highlight the link between post-training and kernel methods. As illustrated in Table 2, using the optimal weights derived from kernel theory immediately a performance boost for the considered network. The post-training step estimate numerically this optimal layer with the gradient descent optimizer. However, computing the optimal weights for the last layer is only achievable for small data set due to the required matrix inversion. Moreover, the closed form solution is known only for specific problems, such as kernelized least square regression. But post-training approaches the same performance in these cases solving (4) with gradient-based methods.

The post-training can be linked very naturally to the idea of pre-training, developed notably by Larochelle et al. (2007), Hinton et al. (2006) and Hinton & Salakhutdinov (2006). The unsupervised pre-training of a layer is designed to find a representation that captures enough information from the data to be able to reconstruct it from its embedding. The goal is thus to find suitable parametrization of the general layers to extract good features, summarizing the data. Conversely, the goal of the post-training is, given a representation, to find the best parametrization of the last layer to discriminate the data. These two steps, in contrast with the usual training, focus on respectively the general or specific layers.

## 6   CONCLUSION

In this work, we studied the concept of post-training, an additional step performed after the regular training, where only the last layer is trained. This step is intended to take fully advantage of the data representation learned by the network. We empirically shown that post-training is computationally inexpensive and provide a non negligible increase of performance on most neural network structures. While we chose to focus on post-training solely the last layer – as it is the most specific layer in the network and the resulting problem is strongly convex under reasonable prerequisites – the relationship between the number of layers frozen in the post-training and the resulting improvements might be an interesting direction for future works.

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

## A    CONVEX LOSS

We show here, for the sake of completeness, that the post-training problem is convex for the softmax activation in the last layer and the cross entropy loss. This result is proved showing that the hessian of the function is positive semidefinite, as it is a diagonally dominant matrix.

**Proposition 2** (convexity). $\forall N, M \in \mathbb{N}, \forall X \in \mathbb{R}^N, \forall j \in \left[1, M\right]$, *the following function F is convex:*

$$F : \mathbb{R}^{N \times M} \mapsto \mathbb{R}$$

$$W \to \log \left( \sum_{i=1}^{M} \exp(XW_i) \right) - \sum_{i=1}^{M} \delta_{ij} \log \left( \exp(XW_i) \right).$$

*where $\delta$ is the Dirac function, and $W_i$ denotes the $i$-th row of a $W$.*

*Proof 1.* Let

$$P_i(W) = \frac{\exp(XW_i)}{\sum_{j=1}^{M} \exp(XW_j)}.$$

then

$$\frac{\partial P_i}{\partial W_{m,n}} = \begin{cases} -x_n P_i(W) P_m(W) & \text{if } i \neq m \\ -x_n P_m^2(W) + x_n P_m(W) & \text{otherwise} \end{cases}$$

Noting that

$$F(W) = -\sum_{i=1}^{M} \delta_{ij} \log \left( P_i(W) \right),$$

we have

$$\frac{\partial F(W)}{\partial W_{m,n}} = -\sum_{i=1}^{M} \delta_{ij} \frac{1}{P_i(W)} \frac{\partial P_i}{\partial W_{m,n}}$$

$$= x_n \left( \sum_{i=1}^{M} \delta_{ij} P_m(W) - \delta_{mj} \right)$$

$$= x_n \left( P_m(W) - \delta_{mj} \right),$$

hence

$$\frac{\partial^2 F(W)}{\partial W_{m,n} \partial W_{p,q}} = x_n \left( \frac{\partial P_m}{\partial W_{p,q}} \right),$$

$$= x_n x_q P_m(W) \left( \delta_{m,p} - P_p(W) \right).$$

Hence the following identity

$$H(F) = \mathbf{P}(W) \otimes (XX^{\mathsf{T}})$$

where $\otimes$ is the Kronecker product, and the matrix $\mathbf{P}(W)$ is defined by $\mathbf{P}_{m,p} = P_m(W) \left( \delta_{m,p} - P_p(W) \right)$. Now since $\forall 1 \leq m \leq M$,

$$\sum_{p=1, p \neq m}^{M} \left| \mathbf{P}_{m,p} \right| = P_m(W) \sum_{p=1, p \neq m}^{M} P_p(W)$$

$$= P_m(W) \left( 1 - P_m(W) \right)$$

$$= \mathbf{P}_{m,m}$$

$\mathbf{P}(W)$ is thus a diagonally dominant matrix. Its diagonal elements are positive

$$\mathbf{P}_{m,m} = P_m(W)\left(1 - P_m(W)\right) \geq 0, \quad \text{as } P_m(W) \in [0,1]$$

and thus $\mathbf{P}(W)$ is positive semidefinite. Since $XX^{\mathsf{T}}$ is positive semidefinite too, their Kronecker product is also positive semidefinite, hence the conclusion.

$\square$

