# OpenReview forum: "Post-training for Deep Learning"
_ICLR.cc/2018/Conference — Reject_

### Official Review · AnonReviewer2 · 2017-11-19
**Overall, this is a nice idea, with experimental results showing its merits. Some of the empirical outcomes do not show that much improvement. However it is a step that could be used either way, even for a slight gain. It would have been more preferable to consider the future directions stated in the paper in this work.**

**Rating:** 5
**Confidence:** 4

**Review:**

Summary:
Based on ideas within the context of kernel theory, the authors consider post-training of NNs as an extra training step, which only optimizes the last layer of the network.
This additional step makes sure that the embedding, or representation, of the data is used in the best possible way for the considered task (which is also reflected in the experiments).

According to the authors, the contributions are the following:
1. Post-training step: keeping the rest of the NN frozen (after training), the method trains the last layer in order to "make sure" that the representation learned is used in the most efficient way.
2. Highlighting connections with kernel techniques and RKHS optimization (like kernel ridge regression).
3. Experimental results.

Clarity:
The paper is well-written, the main ideas well-clarified.

Importance:
While the majority of papers nowadays focuses on the representation part (i.e., how we get to \Phi_{L-1}(x)), this paper assumes this is given and proposes how to optimize the weights in the final step of the algorithm. This by itself is not enough to boost the performance universally (e.g., if \Phi_{L-1} is not well-trained, the problem is deeper than training the last layer); however, it proposes an additional step that can be used in most NN architectures. From that front (i.e., proposing to do something different than simply training a NN), I find the paper interesting, that might attract some attention at the conference.

On the other hand, to my humble opinion, the experimental results do not show a significant gain in the performances of all networks (esp. Figure 3 and Table 1 are within the range of statistical error). In order to state something like this universally, either one needs to perform experiments with more than just MNIST/CIFAR datasets, or even more preferably, prove that the algorithm performs better.

Originality:
It would be great to have some more theory (if any) for the post-training step, or investigate more cases, rather than optimizing only the last layer.

Comments:
1. I assume the authors focused in the last layer of the NN for simplicity, but is there a reason why one might want to focus only on the last layer? One reason is convexity in W of the problem (2). Any other?

2. Have the authors considered (even in practice only) to include training of the last 2 layers of the NN? The authors state this question in the future direction, but it would make the paper more complete to consider it here.

---

### Official Review · AnonReviewer1 · 2017-11-25
**Review of 'Post-Training in Deep Learning'**

**Rating:** 3
**Confidence:** 5

**Review:**

This paper proposes to fine-tune the last layer while keeping the others fixed, after initial end-to-end training, viewing the last layer learning under the light of kernel theory (well actually it's just a linear model).

Summary of evaluation

There is not much novelty in this idea (of optimizing carefully only the last layer as a post-training stage or treating the last layer as kernel machine in a post-processing step), which dates back at least a decade, so the only real contribution would be in the experiments. However the experimental setup is questionable as it does not look like the same care has been given to control overfitting with the 'regular training' method.

More details

Previous work on the same idea: at least a decade old, e.g., Huang and LeCun 2006. See a review of such work in 'Deep Learning using Linear Support Vector Machines' more recently.

Experiments

You should also have a weight norm penalty in the end-to-end ('regular training') case and make sure it is appropriately and separately tuned (not necessarily the same value as for the post-training). Otherwise, the 'improvements' may simply be due to better regularization in one case vs the other, and the experimental curves suggest that interpretation is correct.

---

### Official Review · AnonReviewer3 · 2017-11-29
**This paper demonstrate that by freezing all the penultimate layers at the end of regular training improves generalization.**

**Rating:** 4
**Confidence:** 4

**Review:**

This paper demonstrate that by freezing all the penultimate layers at the end of regular training improves generalization. However, the results do not convince this reviewer to switch to using 'post-training'.

Learning features and then use a classifier such as a softmax or SVM is not new and were actually widely used 10 years ago. However, freezing the layers and continue to train the last layer is of a minor novelty. The results of the paper show a generalization gain in terms of better test time performance, however, it seems like the gain could be due to the \lambda term which is added for post-training but not added for the baseline. c.f. Eq 3 and Eq 4.
Therefore, it's unclear whether the gain in generalization is due to an additional \lambda term or from the post-training training itself.

A way to improve the paper and be more convincing would be to obtain the state-of-the-art results with post-training that's not possible otherwise.

Other notes,

Remark 1: While it is true that dropout would change the feature function, to say that dropout 'should not be' applied, it would be good to support that statement with some experiments.

For table 1, please use decimal points instead of commas.

---

### Public Comment · (anonymous) · 2017-11-17
**Alternative interpretation**

Dear authors,

Thank you to the authors for this interesting paper, which I enjoyed :)

The goal of the proposed method is to separate representation learning (done by the network in all but the final layer) and the task at hand (e.g., classification, which is performed by the final layer of the network). As such, the authors propose a final 'post training' step, which only updates the final layers of a network. One benefit of such an approach is that, given a fixed representation, learning the final layer weights is convex for many choice of the activation functions in the final layer.

The authors propose an interesting link with kernels. From reading the paper, it seemed to me that there was also a relationship between representation learning from the perspective of non-linear ICA, recently proposed by Hyvarinen & Morioka (2016), which I thought I would share with you.
In their work, Hyvarinen & Morioka show that when a classification task is combined with a function approximator (eg a deep net) , the final representation learnt by the network (i.e., what the authors here refer to as $\Phi_{L-1}(x)$) will be equal to the independent components which generated the data (roughly speaking). As a result, it may be possible/interesting to interpret post training as first learning the non-linear unmixing of independent components followed by post training which then performs classification on the original independent components.

Disclaimer: I am not associated with the referenced paper, just wanted to provide an additional justification for the proposed method :)

Good luck!

References:
Hyvarinen & Morioka, "Unsupervised Feature Extraction by Time-Contrastive Learning and Nonlinear ICA", NIPS 2016

---

### Public Comment · ~AhmadReza_GodarzvandChegini1 · 2017-12-16
**ICLR 2018 Reproducibility Challenge**

The paper on Post-Training in Deep Learning suggests another phase of training after a phase of regular training in neural networks. The second phase involves freezing all but the last layer and optimizing the loss function with respect to the weights at the last layer over several additional iterations of training. The authors assert that this additional phase can lead to an improvement of performance in neural networks.

We attempted to reproduce the experiments performed by the authors in order to verify the claims in the paper. The paper is crisp, clear, and well-written in its explanations and hence facilitated the understanding and reproducibility process. In addition, the authors have kindly made their code public, and all experiments were conducted on either well-known datasets or those that could be generated with details provided in the paper. We found that the clarity of the code is also reasonable and any technical details lacking in the report could, for the most part, be found in the code.

The authors performed experiments on three major classes of neural networks, namely CNNs, RNNs, and Feed Forward Neural Networks, using different datasets to compare the performance of additional post-training with classical training. We observed a discrepancy between the paper and its implementation in the setting of these iterations. In general, in the provided code, the accuracy or error comparison (based on the experiment) is made between q iterations of regular training and q regularly-trained iterations + p iterations of post-training. In only the experiment of the CIFAR-10 dataset using CNN, q+100 iterations of regular-training were compared with q regularly-trained iterations + 100 iterations of post-training. Nonetheless, for all observed iterations and network complexities, we confirmed the authors’ findings that applying post-training, on average, improves test accuracy, and executes faster than regular training per iteration. That being said, through our own experimentation, we found that in the Kernel Ridge Regression experiment, an additional number of iterations of regular training would, in-fact, result in lower test error than the equal number of additional iterations of post-training, without necessarily overfitting. For example, at 250 iterations on the fully-connected network using the Parkinson Telemonitoring dataset, the error claimed by the authors in Table 2 of the paper is 0.832. With an additional 200 iterations of post-training, the error reduces to 0.433, however if instead 200 iterations of regular training were performed, we found that the error would have reduced to 0.167. Also, for the CNN experiment on the MNIST dataset, there was no clear relationship between the number of training iterations in the paper and those run in the code, which meant we could not reproduce the training conditions with certainty.

Overall, we believe that the paper is well presented and the experiments support the advantages of post-training stated by the authors. We conclude that the paper is reproducible.

Link to full report: https://github.com/deekshaarya4/Post_training/blob/master/reproducing-post-training-deeplearning.pdf

---

> ### Author Response · Authors · 2017-12-18
> **Reproducibility**
>
> We would like to thank you for taking the time to read the paper and to test the code.
> Your comments will help us improve the paper reproducibility.

---

### Author Response · Authors · 2017-12-18
**Rebuttal**

We would like to thank the reviewers for their feedbacks.

We agree that our paper could benefit from additional experiments, particularly with more recent networks. Additionally, we concur that additional experiments studying the influence of the L2 regularisation on the regular / post training steps could help highlight the interest of the post training step.

However, we would like to point out that we conducted additional experiments and  it is the authors’s belief that while post training is not a very complex or fully original idea, it does seem to provide interesting improvement to the performance of many networks for a negligible cost — and thus worth exploring.

Overall, we acknowledge the reviewer decision to reject this paper and will work to improve it.

---

### Decision · Program_Chairs · 2018-01-29
**ICLR 2018 Conference Acceptance Decision**

**Decision:**

Reject

**Comment:**

* the proposed fine-tuning of only the last layer is not novel enough
* experiments are not sufficient to isolate the differences to support the benefit of post-training